



# Technical note: Characterization of a single-beam gradient force aerosol optical tweezer for droplet trapping, phase transitions monitoring, and morphology studies

Xiangyu Pei[1]&Yikan Meng[1], Yueling Chen[1], Huichao Liu[1], Yao Song[1], Zhengning Xu[1], Fei Zhang[1], Thomas C. Preston[3], Zhibin Wang[1,2*]

[1]Zhejiang Provincial Key Laboratory of Organic Pollution Process and Control, MOE Key Laboratory of Environment Remediation and Ecological Health, College of Environmental and Resource Sciences, Zhejiang University, Hangzhou 310058, China

[2]ZJU-Hangzhou Global Scientific and Technological Innovation Center, Zhejiang University, Hangzhou 311215, China

[3]Department of Atmospheric and Oceanic Sciences and Department of Chemistry, McGill University, 805 Sherbrooke Street West, Montréal, Quebec H3A 0B9, Canada

*Correspondence to*: Zhibin Wang (wangzhibin@zju.edu.cn)

Xiangyu Pei and Yikan Meng contribute equally to this work.

**Abstract.** Single particle analysis is essential for a better understanding of the particle transformation process and predicting its environmental impact. In this study, we developed an aerosol optical tweezer (AOT)-Raman spectroscopy system to investigate the phase state and morphology of suspended aerosol droplets in real time. The system comprises four modules: optical trapping, reaction, illumination and imaging, as well as detection. The optical trapping module utilizes a 532 nm laser and a 100x oil immersion objective to stably trap aerosol droplets within 30 seconds. The reaction module allows us to adjust relative humidity (RH) and introduce reaction gases into the droplet levitation chamber, facilitating experiments to study liquid-liquid phase transitions. The illumination and imaging module employs a high-speed camera to monitor the trapped droplets, while the detector module records Raman scattering light. We trapped sodium chloride (NaCl) and 3-methyl glutaric acid (3-MGA) mixed droplets to examine RH-dependent morphology changes. Liquid-liquid phase separation (LLPS) occurred when RH was decreased. Additionally, we introduced ozone and limonene/α-pinene to generate secondary organic aerosol (SOA) particles in situ, which collided with the trapped droplet and dissolve in it. To determine the trapped droplet's characteristics, we utilized an open-source program which based on Mie theory to retrieve diameter and refractive index from the observed whispering gallery modes (WGMs) in Raman spectra. It is found that mixed droplets formed core-shell morphology when RH was decreased, and the RH dependence of the droplets phase transitions generated by different SOA precursors varied. Our AOT system serves as an essential experimental platform for in-situ assessment of morphology and phase state during dynamic atmospheric processes.



## 1 Introduction

Atmospheric aerosol particles can absorb and reflect solar radiation, be activated into cloud droplets, participate in ice nucleation processes, and provide interfaces for chemical reactions (Mülmenstädt et al., 2015; Shrivastava et al., 2017). As a result, aerosols play important roles in air pollution, atmospheric chemistry and climate change (Pöschl, 2005). Aerosol particles can have complex compositions including inorganic, metallic, and mineral components, and elemental carbon, organic carbon, as well as an amount of water (Kolb and Worsnop, 2012). Aerosol particles can also have different morphologies. For example, aerosol particles composed of inorganic salts and organic components can have solid, partially engulfed or core-shell, and homogeneous morphologies through phase transitions (Freedman, 2020). Changes in aerosol composition and water content will lead to the evolution of particle morphology and phase state, while altering other physiochemical properties such as pH (Freedman et al., 2019), polarity (Zuend and Seinfeld, 2012), interfacial tensions (Sullivan et al., 2018), and photochemistry (Cremer et al., 2016).

To study the physiochemical properties of atmospheric aerosol particles, various measurement methods and techniques are applied. However, traditional measurement methods mainly represent the average properties of the aerosol population, lacking detailed information about individual particles, such as morphology, phase state, composition, and so on. In comparison to methods measuring properties of particle populations, single particle levitation techniques have been developed and applied as effective tools for measuring the physical and optical properties of micron-sized particles (Krieger et al., 2012). Several single particle levitation techniques, such as the electrodynamic balance (EDB), acoustic trap (AT), and optical tweezers (OT), have been widely used to control and conduct direct real-time in-situ measurements of single particles (Davies, 2019; Combe and Donaldson, 2017; Gong et al., 2018b). Using these techniques, various atmospheric aerosol properties during dynamic processes, such as hygroscopicity, volatility, optical properties, viscosity, surface tension, and diffusion characteristics, have been extensively studied (Chan et al., 2005; Davies et al., 2013; Cai et al., 2015). However, the fundamental principles of these techniques determine their applicability for trapping particles of different sizes. For instance, EDB and AT can trap particles with diameter ranges of 5-50 μm and 20-100 μm, respectively, while OT can trap smaller particles with size ranges in the microns and tens of microns in diameter (Krieger et al., 2012).

When transparent or weakly absorptive spherical particles, such as droplets, are trapped by Optical Tweezers (OT) and measured with Raman spectroscopy (RS), these spherical droplets can function as high-finesse optical cavities. This results in a significant enhancement of stimulated Raman scattering signals at specific wavelengths, which are referred to as whispering gallery modes (WGM) (Ashkin and Dziedzic, 1981). The diameter and refractive index of the trapped droplet can be determined from the WGM signals (Reid et al., 2007). Additionally, different droplet morphologies can exert a considerable influence on WGM behavior. For instance, if the droplet exhibits a homogeneous or concentric core-shell morphology, WGM signals are generated. However, when liquid-liquid phase separation (LLPS) occurs, and the droplet assumes a partially





engulfed morphology, WGM signals will vanish. By combining OT with RS, precise information about droplet size, refractive
index, and morphology can be obtained. For example, Rickards et al. (2013) employed OT-RS to investigate the evolution of
size and refractive index under varying relative humidity (RH) conditions, exploring the impact of the O/C ratio on aerosol
hygroscopicity. Gorkowski et al. (2020) utilized OT-RS to predict phase separation and changes in particle morphology. They
studied the mixing behavior of α-pinene secondary organic aerosol (SOA) with different organic phases, including squalene
and glycerol, at various relative humidity levels. Boyer et al. (2020) applied OT-RS to achieve highly accurate in-situ pH
measurements of $NaHSO_4$ microdroplets.
In this study, we present our custom-made aerosol optical tweezer (AOT) system. We describe the system and the design of
the droplet particle levitation chamber. We have innovated the chamber design by adopting a smaller chamber to reduce droplet
capture time. Additionally, we have introduced a double-floor chamber room and a replaceable intermediate plate, which
facilitates control of the flow exchange rate for different experiments. We have also established methods for Raman spectrum
analysis and droplet morphometry determination and have systematically characterized this system, presenting the results. The
application of this system includes studying the morphology of aqueous droplet-hydrocarbon experiments and in-situ
generation and addition of SOA as the second phase. Furthermore, our chamber design provides the possibility for conducting
more gas-liquid phase reaction experiments in the future. This indicates that the AOT system is a powerful tool that can be
used to uncover the mechanisms of changes in physical and chemical properties of droplets during their evolution under
different conditions.
**2 Materials and methods**
**2.1 System description**
A schematic illustration of the aerosol optical tweezer system is presented in Figure 1, which includes the optical trapping
module, illumination and imaging module, detector module, and reaction module. The optical trapping module utilizes a 532
nm laser (Opus 532-2W). Depending on the desired particle size and flow turbulence, the laser power is adjusted between 30
and 200 mW to maintain steady trapping. Subsequently, the laser passes through expansion lenses to overfill the back aperture
of the microscope objective. A 100x oil immersion objective (Olympus, UPLFLN100XO, NA 1.30) is brought into contact
with a glass coverslip (Nest, thickness 160-190 μm) installed at the bottom of the aerosol particle levitation chamber. This
convergence of the laser beam above the glass coverslip forms an optical trap.
The illumination and imaging module consist of a 450 nm LED (Daheng Optics, GCI060404) and a camera (Thorlabs,
CS165CU/M) used for illuminating and imaging the particles. Both the camera and spectrograph capture the LED scattered
light and the Raman scattered light from the particle, respectively, using the same microscope objective. To obtain a clear



image of the particle, a low-pass filter (Andover, 500FL07-25) is positioned in front of the camera lens to eliminate the
influence of backscattered light from the 532 nm laser.
The Raman scattering signals are collected through a detector module, with the primary component being a spectrometer. The
Raman scattered light passes through two 50:50 beam splitters (CVI Laser Optics, BTF-VIS-50-2501M-C) and a notch filter
(Edmund, 86125) before being focused into the Raman spectrograph. A spectrometer (ZOLIX, Omni-λ5004i) is employed to
measure the Stokes-shifted Raman spectra, utilizing a 20 µm entrance slit width and a 1200 groove/mm diffraction grating
with a blaze wavelength of 500 nm to achieve a spectral resolution of 0.021 nm. The wavelength position of the spectrograph
is calibrated using a Hg-light source. For liquid-liquid phase separation experiments, the center wavelengths of 645 nm for the
diffraction grating are configured, and the Raman scattered light is recorded every 4 seconds within a wavelength range of
624.24-665.40 nm.
The reaction module serves as the area where RH airflow and reactive gases are introduced, and where chemical reactions
occur. Its central component is a custom-made aerosol particle levitation chamber (Figure 1(b)), which integrates the inverted
oil immersion objective from the optical trapping module and the downward brightfield illumination source from the
illumination and imaging module. Further details on the chamber's design will be provided in the following section. For
controlling RH airflow, two air streams are combined: one with a relative humidity of 100% and the other air-dried using a
silica drier. This mixture results in conditioned air with a specific RH, allowing precise control over humidity within the
chamber. The humidification of the airflow is achieved through a water bubbler, and both the humidified and dry airflows are
regulated by two mass flow controllers (MFCs) with a total flow rate of 0.3 L/min. Temperature and humidity sensors
(Sensirion, SHT85) are employed to measure the temperature and humidity of the airflow at both the inlet and outlet of the
chamber, following the design used by Gorkowski et al. (2016). Measuring temperature and humidity directly inside the
chamber with a probe is avoided because particle deposition on the probe can affect measurement accuracy. Additionally,
placing the probe near the droplet could interfere with droplet capture, making it challenging to maintain stable trapping of the
droplet. Other reactive gases can be introduced into the chamber through dedicated ports located on the chamber wall for
chemical reactions, and these gases exit through an exhaust port.




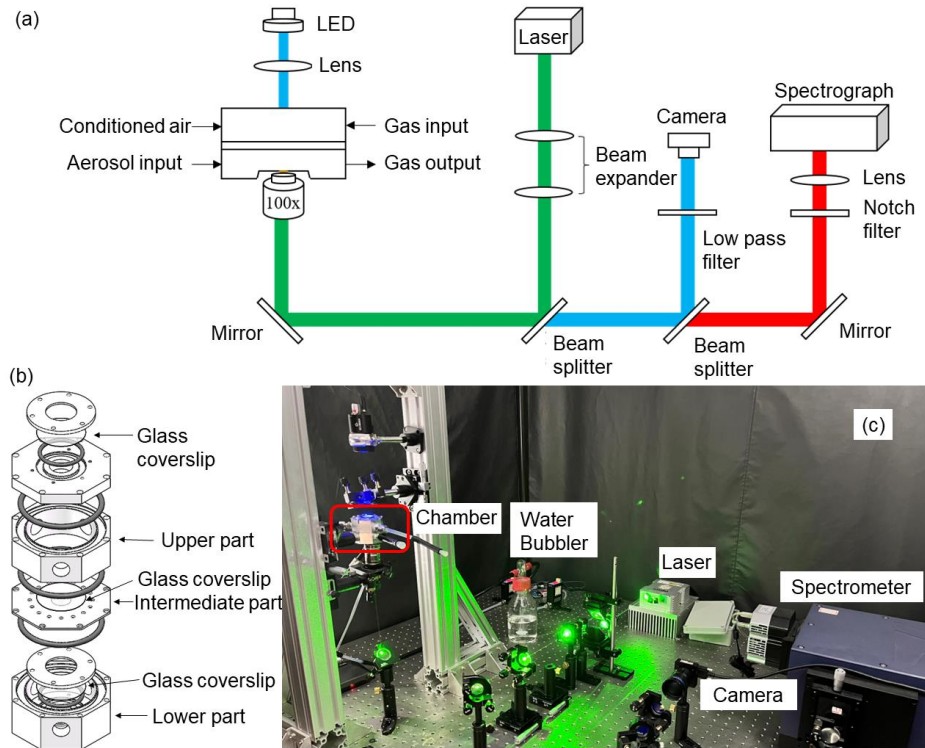

**Figure 1. (a) Schematic of the aerosol optical tweezer setup used in this study. (b) Design of the droplet particle levitation chamber.**

**(c) Photo of the main components of the system, including the chamber, water bubbler, laser, camera, and spectrometer.**

## 2.2 Chamber design

The chamber is designed to provide a sealed environment and to regulate the ambient humidity for the optically trapped droplet,

similar to the chamber design by Gómez Castaño et al. (2019). Our chamber is relatively compact, with a volume of 36 mL,

in contrast to the design by Gorkowski et al. (2016). This smaller size offers several advantages, including improved capture

efficiency. Moreover, we have introduced an innovative double-floor chamber structure. The chamber consists of three main

components: the upper section for introducing RH conditioned airflow and organic aerosol/gas flow, an intermediate plate that

connects the upper and lower compartments for airflow passage, and the lower part for injecting atomized droplets. Both the

upper and lower sections feature three ports for tubing connections. The replaceable intermediate part plays a pivotal role in

enhancing the chamber's versatility. Depending on the specific experimental objectives, the shape, size, and distribution of

ventilation holes on the intermediate part can be customized to control the flow exchange rate. For instance, in experiments

involving liquid-liquid phase separation, a flat plate with unobstructed circular holes in the middle is employed to minimize



the airflow's impact on liquid droplets. In contrast, during reactivity experiments, the central circular hole is altered from a flat
type to a circular barrier type, facilitating better contact between the reactants and the droplet's surface.

To facilitate the passage of light while preventing the flow of gas, a transparent glass slide is positioned on top of the
intermediate section to cover the central holes. This innovative approach allows the illumination LED's light to pass through
the window, reaching the trapped particle, while enabling the gas flow to penetrate the intermediate plate through the holes.
This design effectively reduces the likelihood of disturbing the stability of particle trapping. In the lower section, the vertical
distance between the droplet inlet and the coverslip is set at 6.5 mm. This configuration ensures that atomized droplets can
reach the optical trap position in abundance, thereby increasing the probability of successful trapping. Inside the lower part, a
glass coverslip is placed and soaked in a surfactant solution (a 50:50 water-to-Decon 90 solution) to prevent droplet deposition.
The openings of the chamber are sealed with O-rings and screws to ensure airtightness. This sealing is applied both between
the upper, intermediate, and lower sections, as well as between the window cover, coverslip, and the bottom of the lower part.
This design allows for easy disassembly and cleaning of the entire chamber. Under conditions devoid of external disturbances,
we are able to stably capture droplets within the chamber for periods exceeding 24 hours.
**2.3 Aerosol generation**
Aqueous aerosol droplets are created using a medical nebulizer (LANDWIND, PN 100) and can be effectively trapped within
30 seconds after introducing the aerosol plume into the chamber. In this study, aqueous NaCl droplets were generated to
investigate their response to changes in RH and to assess the accuracy of particle size measurements during the droplet
equilibrium experiments. Additionally, mixed droplets containing both NaCl and 3-methyl glutaric acid (3-MGA) were
generated to demonstrate the research approach for studying liquid-liquid phase separation (LLPS). Detailed information on
each of these experiments is provided below.
Following the successful trapping of droplets in the lower layer of the AOT, reaction gases such as ozone and volatile organic
compounds (VOC) can be introduced into the chamber. Subsequently, SOA is formed and added to the surface of the droplets
through designated ports within the chamber. In this study, both limonene and α-pinene were used as separate SOA precursors.
The concentration of limonene within the AOT was controlled by passing dry nitrogen flow over limonene contained in a
diffusion vial submerged in a temperature-regulated bath system (Pei et al., 2018). Ozone was generated by passing zero air
through a UV lamp unit (SOG-2, UVP). The VOC and ozone reacted in the upper part of the chamber, resulting in the
production of SOA particles. These particles settled and subsequently collided with the trapped droplets in the lower section
of the chamber.



**2.4 Detection of the morphology of mixed droplets**


Droplets exhibit various morphologies in the atmosphere, including homogeneous, partially engulfed, and core-shell
morphologies (Song et al., 2013; Veghte et al., 2013). Whispering gallery modes (WGMs), which significantly enhance Raman
signals (Ashkin and Dziedzic, 1981), serve as crucial indicators for identifying droplet morphology (Gorkowski et al., 2016;
Stewart et al., 2015). These WGMs are observed as distinct peaks superimposed on the broader vibrational modes in the
droplet's Raman spectrum. Analyzing the Raman spectrum involves pinpointing the wavelength positions of WGM peaks in
the spectrum and fitting these positions to a Mie scattering model. This fitting process allows for the retrieval of the droplet's
diameter, denoted as $D_p$, and its refractive index, represented as $n$ (Preston and Reid, 2013, 2015).
Our method for retrieving the values of $D_p$, and $n$ from the WGMs comprises two essential components: an automatic peak
finding algorithm and a Mie scattering fitting program. The peak finding method relies on the ipeak code, which was developed
by (O'haver, 2022). This approach accurately identifies the desired peaks by smoothing the first derivative of the signal and
identifying downward-going zero-crossings that meet specific predefined criteria, such as minimum slope and amplitude
thresholds. Once the positions of the WGMs are determined using the peak finding method, we employ the Mie scattering
fitting program known as Mie Resonance Fitting (MRFIT), developed by Preston and Reid (2015). MRFIT is utilized to
calculate both the diameter and refractive index of a homogeneous droplet. It provides mode assignment information, including
the mode number, mode order, and polarization, which is essential for a comprehensive analysis.
During the experiment, typically, a homogeneous droplet is initially trapped. Subsequently, as the RH is decreased, the droplet
may undergo phase separation and transform into partially engulfed or core-shell morphologies. These transformations have
distinct effects on the WGMs. When a droplet transitions into a partially engulfed state, its symmetric structure is disrupted,
leading to quenching of the WGMs. In contrast, when the droplet assumes a core-shell structure, the WGMs weaken because
the radial uniformity of the droplet is perturbed (Buajarern et al., 2007; Mitchem et al., 2006). Consequently, applying MRFIT
to a partially engulfed or core-shell droplet can render the retrieval of diameter and refractive index implausible, resulting in
abnormally high fit errors. To address this issue and retrieve the diameters and refractive indices for core-shell droplets, we
employ another program called Mie Resonance Shell Fitting (MRSFIT), developed by . MRSFIT is specifically designed to
fit observed Mie resonances to the resonances predicted using Mie theory for core-shell particles. The mode assignments
provided by MRFIT guide the selection of appropriate parameters for core-shell droplets. After capturing a droplet, its
morphology can be identified from the spectra, with examples illustrated in Figure 2.



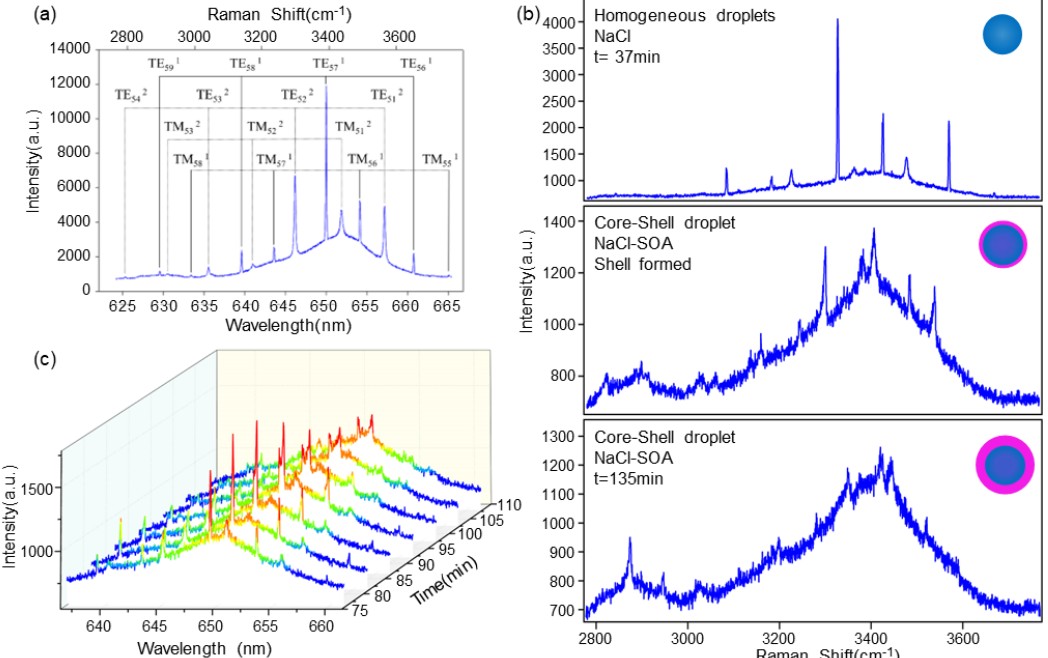

**Figure 2. Example of identifying droplet morphology based on spectral features. (a) Example of a Raman scattering signature from a trapped aqueous NaCl droplet. (b) Spectra of different droplet morphologies: Upper subgraph shows the typical spectrum of homogeneous aqueous saturated NaCl droplets. Middle subgraph shows the spectrum when SOA forms a thin shell on the surface of the saturated NaCl droplets. Bottom subgraph shows the spectrum with weakened WGMs peaks when SOA continues to coagulate onto the surface of the saturated NaCl droplets. (c) Example of WGM splitting time series: red peaks gradually split from one into two, and intensity becomes weaker when SOA is added to the droplet, indicating the formation of a core-shell morphology.**

## 3 Results and discussion

### 3.1 Performance of trapping chamber

In the initial stages of chamber optimization, a 3D printing technique was employed to create the chamber prototype. However, for the final chamber design, aluminium was chosen as the construction material instead of 3D printing material. While 3D printing offers rapid prototyping capabilities, the polymer structure of 3D printing materials can absorb moisture, making it impractical to maintain a stable RH level within the chamber (Gorkowski et al., 2016). Initially, a single-floor chamber design was used to achieve successful particle trapping. However, it was challenging to introduce controlled RH flow into the chamber while maintaining stable droplet trapping. This difficulty arose because the ports on the chamber were located on the sides, and the flows were delivered directly to the trapping position, thereby disrupting the stability of trapping. Consequently, a double-floor chamber design was adopted. In this configuration, RH flow is introduced into the upper part of the chamber and then directed through holes in the intermediate part to reach the lower section. All the experimental data presented in this study



were obtained using the double-floor chamber design, which offered improved control over RH conditions and allowed for
stable droplet trapping.
A saturated aqueous NaCl droplet was trapped to investigate its response to changes in RH. These droplets equilibrate rapidly
with variations in the RH of the surrounding air (Gorkowski et al., 2016). When the droplet was successfully trapped, the RH
was systematically ramped up and down in approximately 4% steps. This RH cycling ranged from 74% to 98%, and the settings
for RH, including the flows of both humid air and dry air, were held constant for 20 minutes at each step. This process of
ramping RH up and down was repeated seven times, totaling 31 hours of experimentation. Measurements of RH both before
and after the chamber were taken. Simultaneously, Raman scattered light was recorded at intervals of 4 seconds, enabling the
calculation of $D_p$ and $n$ through the use of the WGM fitting program known as MRFIT. In Figure 3, the retrieved values for
$D_p$ and $n$, as well as the measured RH before and after the chamber, are displayed. Additionally, the figure provides a time
series of Raman spectra for a trapped aqueous NaCl droplet during the first RH cycling experiment. These data offer insights
into how the droplet responds to RH variations.

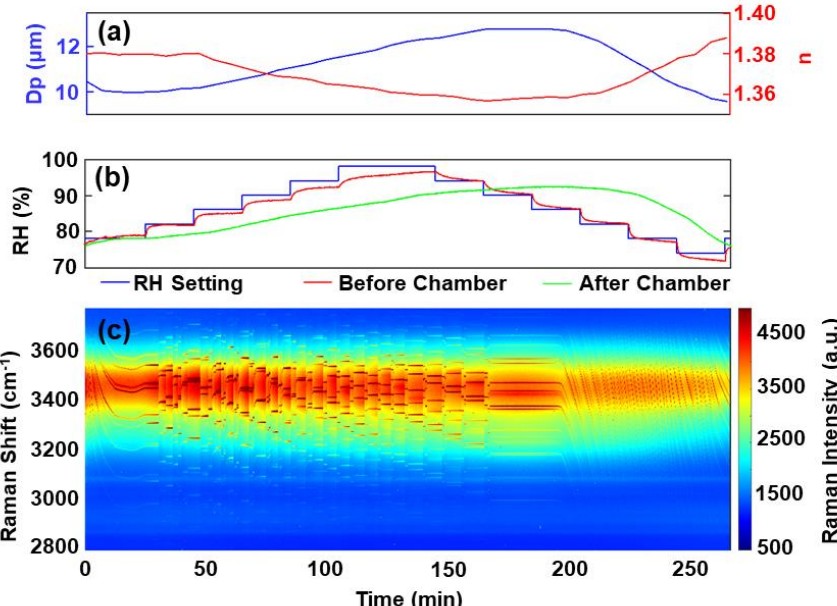

**Figure 3. (a) Retrieved diameter (Dp) and refractive index (n). (b) RH measured before and after the chamber. (c) Time series of**
**Raman spectra for a trapped aqueous NaCl droplet.**

Figure 3(b) clearly illustrates a significant difference between the RH measurements before and after the chamber. Specifically,
the RH measured before entering the chamber closely followed the stepwise setting values, while the RH measured after





exiting the chamber exhibited a continuous change with a noticeable lag compared to the RH before entering. When the RH
was incrementally increased, the lag between the RH before entering the chamber and the RH after exiting the chamber was
20 minutes for an RH of 78% and extended to 80 minutes for an RH of 92%. Conversely, when the RH was decreased, the lag
between the RH before entering the chamber and the RH after exiting the chamber was 42 minutes for an RH of 90% and
reduced to 20 minutes for an RH of 77%. This observed lag can be attributed to the fact that water vapor can only traverse
through the holes in the intermediate part from the upper section to the lower section, which necessitates time for the RH in
the lower part to reach the same value as in the upper part. The lag in RH reflects the time required for the chamber's internal
conditions to equilibrate with the externally controlled RH levels.
Figure 3(c) displays the Raman spectra time series for the trapped aqueous NaCl droplet. This spectrum exhibits the broad O-
H vibration mode from water, spanning the range of 3200-3600 cm$^{-1}$, with sharp WGMs superimposed on top, highlighted in
dark red. In Figure 3(a), we observe the retrieved values for $D_p$ and $n$ of the trapped droplet. During the initial 10 minutes of
the experiment, the WGM positions blue-shifted to shorter wavelengths (as shown in Figure 3(c)), and the droplet's diameter
decreased from 10.47 μm to 9.98 μm. This indicates that the newly nebulized and trapped droplet did not immediately
equilibrate with the surrounding air, and water was evaporating from the droplet. Starting from 25 minutes into the experiment,
as RH was increased, there was a rapid shift in WGM positions to longer wavelengths within 1 minute. After this initial shift,
the positions remained relatively stable, but the intensities of the WGMs increased significantly over the rest of the time.
Additionally, there were instances where the WGM positions shifted again abruptly, but the red shift phenomenon was not
always observed. Nevertheless, an increase in WGM intensities did occur. This process continued until the RH after exiting
the chamber reached its maximum value of approximately 92% and remained stable from 160 minutes to 195 minutes.
Subsequently, during the period from 195 minutes to 267 minutes, as RH was decreased, the RH after exiting the chamber
gradually declined from 92% to 76%. Interestingly, unlike during the RH increase, there were no abrupt changes in WGMs.
Instead, the WGMs consistently shifted to shorter wavelengths, indicating continuous water vapor evaporation from the droplet.
This resulted in a decrease in droplet diameter. In general, the trend of $D_p$ correlated well with the RH after exiting the chamber,
increasing from 10.0 μm to 12.8 μm as RH increased from 78% to 92%, and decreasing from 12.8 μm to 9.6 μm as RH
decreased from 92% to 76%. This suggests that the droplet responded quickly to changes in its surrounding RH. Regarding $n$,
it exhibited a reverse trend compared to $D_p$. It decreased from 1.379 to 1.357 as RH increased from 78% to 92%, and increased
from 1.357 to 1.388 as RH decreased from 92% to 76%. This indicates that as RH increased, more water molecules were added
to the droplet, diluting the NaCl solution and causing the refractive index to approach that of pure water (~1.33). This trend
aligns with previous studies (Boyer et al., 2020) and demonstrates the effectiveness of the Mie scattering fitting program,
MRFIT, developed by Preston and Reid (2015), in providing reasonable and consistent results.



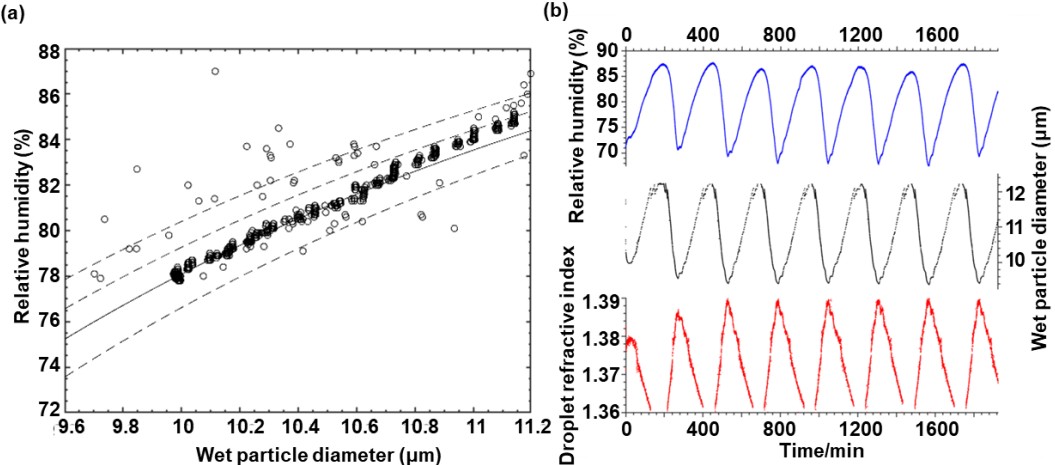


**Figure 4. Chamber humidity performance. (a) Comparison between experimental measurements and theoretical calculations of the**
**relationship between wet particle diameter and relative humidity. The lines represent Köhler curves calculated for particles with**
**different diameters: 5.28 μm, 5.38 μm, 5.48 μm, and 5.60 μm from left to right, respectively. The points represent the results obtained**
**using our AOT to observe particles with a 5.48 μm diameter, which is consistent with the predicted result (black solid line). (b) Time**
**series of RH, droplet diameter, and refractive index variation in the chamber.**

263

To assess the accuracy of particle size measurements using the RH balance, the method proposed by Mitchel et al. (2006b)

was employed. This method estimates the particle size of a single droplet by comparing the actual size of the droplet during

equilibrium with the theoretical value calculated using Köhler's theory. In this specific experiment, droplets were generated

using a 0.4 g/mL NaCl aqueous solution, and the results are presented in Figure 4(a). Upon reaching initial equilibrium within

the chamber, the size of the droplet, which initially had a diameter of 9.6 μm, exhibited excellent consistency with the predicted

results based on Köhler's theory for a dry particle with a diameter of 5.48 μm. Figure 4(b) illustrates the variations in droplet

diameter and refractive index within the RH range of 68% to 88% over a span of 32 hours during the droplet equilibrium

experiment. Throughout the process of increasing and decreasing RH, the droplet diameter exhibited a direct proportionality

to RH, while the refractive index displayed an inverse proportionality to RH. These trends highlight how changes in RH

influence the droplet's size and optical properties.

**3.2 Phase separation of inorganic/ SOA proxy mixed aqueous droplets**

Droplets were generated using a medical nebulizer (LANDWIND, PN 100) and composed of a mixed saturated solution of

NaCl and trimethyl glutaric acid (3-MGA). The solution had a mass concentration of 100g/L, and the organic-to-inorganic

mass ratio was maintained at 1:1. These droplets, with diameters ranging from 8-12 μm, were subsequently captured using the



aerosol optical tweezers. Inside the chamber, the RH was adjusted while Raman spectra were recorded. This allowed for the
monitoring of changes in the droplet's morphology, following the method described in Section 2.4 of the study. The recorded
Raman spectra provided insights into how alterations in RH affected the morphological characteristics of the droplets.

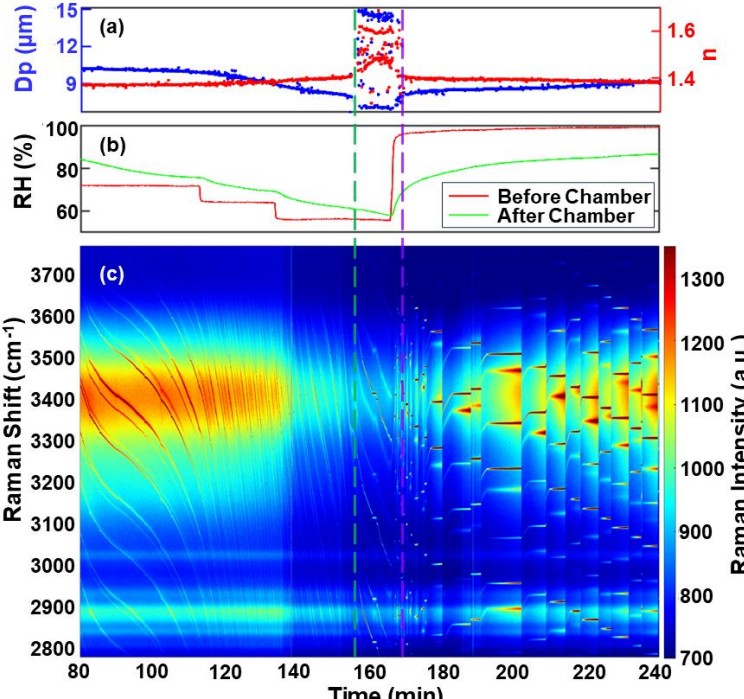


**Figure 5. Liquid-liquid phase separation and mixing of a NaCl/3-MGA mixed solution. (a) The droplet diameter and refractive index**
**obtained from WGM fitting, with blue dots representing the droplet diameter and red dots representing the refractive index. (b)**
**The change in RH of the chamber, with red lines representing RH before entering the chamber and green lines representing RH**
**exiting the chamber. (c) Time-resolved Raman spectra, with WGMs marked in dark red. The green dashed line and purple dashed**
**line represent the occurrence of liquid-liquid phase separation and liquid-liquid phase mixing, respectively.**

Figure 5 illustrates an experimental example involving the liquid-liquid phase separation and mixing of a NaCl/3-MGA mixed
solution droplet. The experiment began with the chamber's RH being stabilized at 95% for a duration of 20 minutes. During
this phase, the initial droplet diameter was determined to be 11.38 μm, and its refractive index was measured at 1.360.
Subsequently, a RH cycling process was initiated, involving a gradual decrease and then increase in the RH inside the chamber.
During the RH decrease phase, the WGMs shifted towards shorter wavelengths, and the WGM locations exhibited a negative
slope. These observations indicated that the droplet's diameter decreased due to the evaporation of water from within the
droplet. This decrease in diameter resulted in an increase in the solute concentration within the droplet, causing the refractive
index of the droplet to rise. At the 155-minute mark, the WGMs in the Raman spectrum became weakened but did not disappear





297 entirely. Meanwhile, the fitting errors associated with the determination of droplet diameter and refractive index significantly

298 increased. These findings suggested that LLPS had occurred, and a core-shell morphology was formed within the droplet. The

299 separation RH (SRH) for the NaCl/3-MGA mixture was determined to be 61.5%. As the RH was increased during the

300 subsequent phase, the WGMs shifted towards longer wavelengths. When the RH reached 65.5%, the errors associated with the

301 WGMs fitting algorithm returned to the state observed before phase separation, indicating the restoration of a homogeneous

302 state within the droplet. Therefore, the RH level of 65.5% was considered to be the mixed RH (MRH) corresponding to the

303 conditions of the droplet during this experiment.

304 **3.3 Morphology of NaCl droplet coated with SOA**

305 To investigate the morphology of inorganic droplets coated with SOA, SOA precursors such as limonene and α-pinene were

306 oxidized with ozone inside the chamber, generating SOA in situ. Raman-enhanced spectroscopy was used to determine the

307 droplet morphology, revealing that SOA formed a second phase and exhibited a tendency to create a shell on the surface of

308 aqueous droplets. Figure 6 presents an experiment involving droplets coated with limonene SOA. At the start of the experiment,

309 a saturated NaCl droplet was trapped at 0 minutes. The pink stripe in the figure represents the introduction of ozone and

310 limonene into the chamber to generate SOA, with a RH of approximately 73.10±0.18%. The median particle size of the

311 generated SOA particles was measured as 25.67 nm using a scanning mobility particle size (SMPS) instrument. As SOA was

312 introduced into the chamber, the WGMs became weakened but did not completely disappear. Over time, with the continuous

313 introduction of SOA, the SOA shell gradually formed and thickened, disrupting the radial homogeneity of the droplets and

314 leading to the appearance of two phases within the droplet. Consequently, one WGM peak began to split into two peaks (as

315 observed in Figure 2(c) and Figure 6(b)), and the fitting error of the homogeneous Mie algorithm increased (as shown in Figure

316 6(a)). These changes indicated the formation of a core-shell morphology within the droplet. As the organic component content

317 increased due to the presence of SOA, the spontaneous organic peaks in the spectra (in the 2800-2900 cm$^{-1}$ region) were

318 enhanced. WGMs also emerged in the C-H hydrocarbon region, although they weakened in the OH region. Over time, the

319 intensity of the WGMs peak in the C-H region continued to increase when SOA was continually added, as depicted in Figure

320 6(c). These observations provide insights into the evolving morphology and composition of the droplets as SOA is introduced.

321





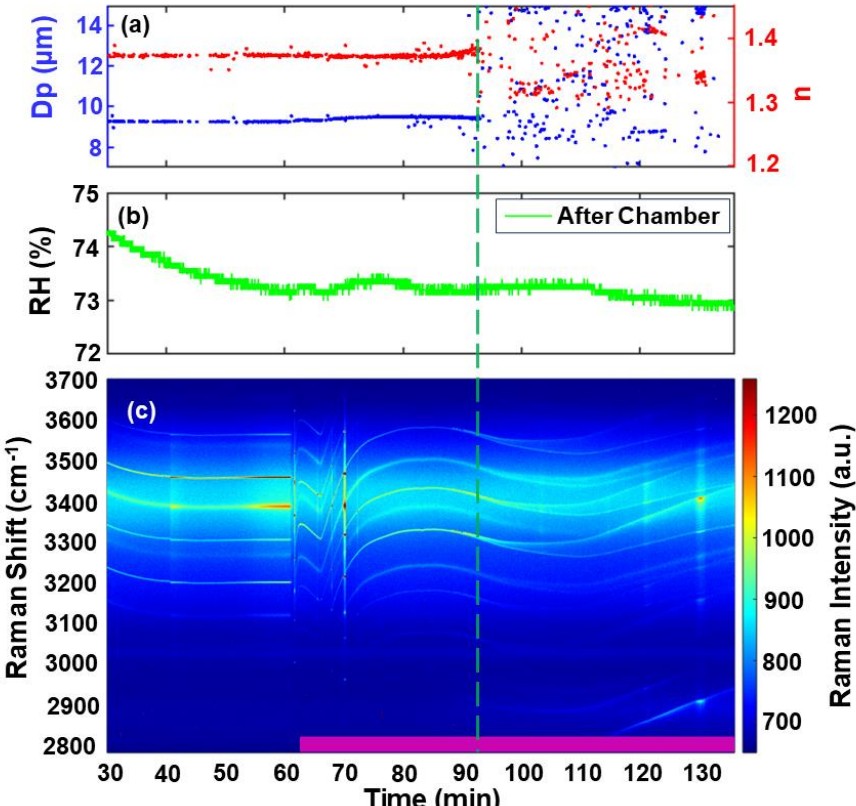

**Figure 6. The experiment of limonene SOA coated on a saturated NaCl droplet. (a) Retrieved diameter (blue dots) and refractive index (red dots) of the droplet. (b) Relative humidity (RH) of the flow after exiting the chamber. (c) Limonene SOA (purple bar at the bottom) was added to the droplet, resulting in the formation of a core-shell morphology. The green dashed line indicates the occurrence of WGM splitting and the formation of a second phase.**



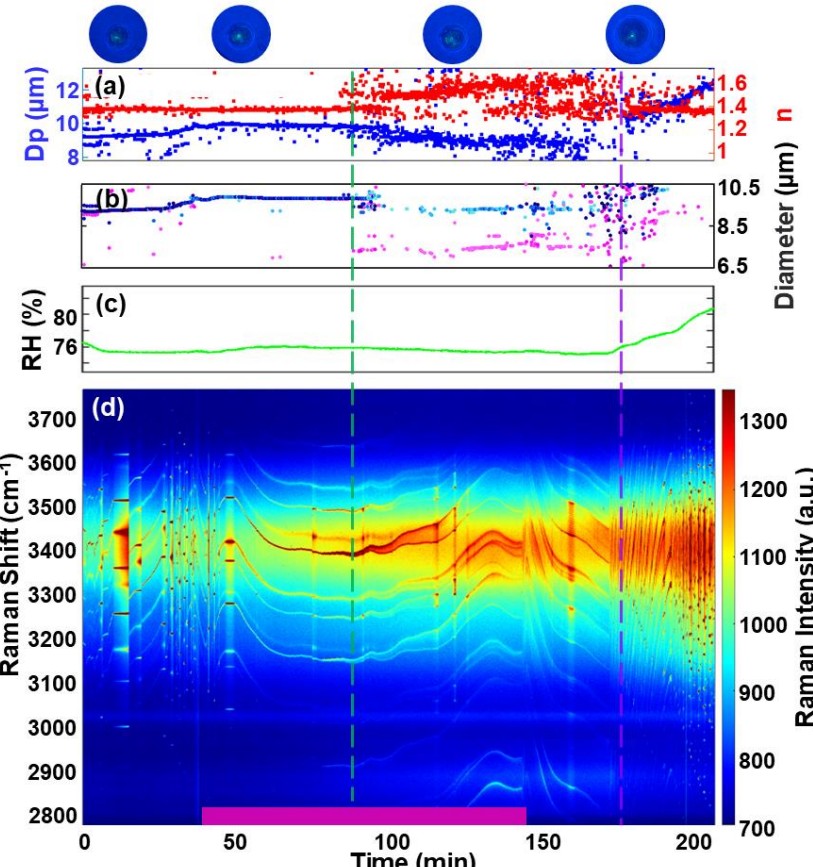

**Figure 7. The experiment of α-pinene SOA coated on a saturated NaCl droplet. (a) Retrieved diameter (blue dots) and refractive index (red dots) of the droplets using the homogeneous droplet model, with real-time images of the droplet at different times. (b) Retrieved shell diameter (blue dots) and core diameter (red dots) of droplets using the core-shell droplet model. The darker the color, the smaller the fitting error. Between the green dotted line and purple dotted line, blue dots represent shell diameter, while pink dots represent core diameter. (c) RH of the flow after exiting the chamber. (d) Limonene SOA (purple bar at the bottom) was added to the droplet, resulting in the formation of a core-shell morphology. The green dashed line and purple dashed line represent the occurrence of liquid-liquid phase separation and liquid-liquid phase mixing, respectively.**

Figure 7 presents another experiment involving a droplet coated with α-pinene SOA at a RH of approximately 75.47±0.29%. Using the same SOA generation conditions, the median diameter of the α-pinene SOA particles was measured at 25.67 nm. After approximately 40 minutes of introducing SOA into the chamber, significant changes occurred in the WGMs, including splitting, and the fitting error of the homogeneous Mie algorithm increased significantly. As SOA was generated, ultra-micron SOA particles formed within the NaCl droplet. This phenomenon can be attributed to the higher polarity of certain compounds in SOA, which are water-soluble and can dissolve into the aqueous phase (Gong et al., 2018a; Takeuchi et al., 2022; Mutzel et al., 2021). This behavior has been observed in previous studies using electron microscopy, where the formation of outer





shell emulsion droplets in organic/inorganic mixed droplets was observed (Song et al., 2012). Similar phenomena have also
been observed in experiments involving AOT (Gorkowski et al., 2017). With continued introduction of SOA, a shell gradually
formed on the surface of the aqueous phase of the droplet. To illustrate the development of this shell, a core-shell model
developed by Vennes and Preston (2019) was used to calculate the inner core and outer shell diameters of the droplet during
the phase addition stage. Eventually, a shell with a thickness of approximately 100 nm was formed (as shown in Figure 7(b)).
Some gaps in the data are due to the WGMs' insufficient quality during this stage, which affected the fitting of the core-shell
model. The automated peak finding program used may not have recognized WGMs with slightly weaker intensity. Figure 7(b)
reveals that before 88 minutes, the core-shell model provided identical values for the fitted core and shell diameters, indicating
a homogeneous droplet with no shell. However, after 88 minutes, the difference between the core and shell diameters increased,
signifying the development of a core-shell structure within the droplet.
Interestingly, the formation of a shell was observed at an RH of 75.47% when α-pinene was used to generate SOA. However,
under the same conditions, no shell formation was observed when limonene was used to generate SOA. However, when the
RH was reduced to 73.10%, shell formation was observed with limonene SOA. This difference in behavior can be attributed
to the chemical properties of the SOA precursors. Limonene contains two double bonds, making it more reactive to ozone and
resulting in a higher yield of SOA compared to α-pinene (Chen and Hopke, 2010; Saathoff et al., 2009). Consequently, SOA
generated from α-pinene has lower water solubility in saturated NaCl aqueous solution, which explains the different phase
states observed in NaCl droplets mixed with α-pinene SOA versus limonene SOA. After introducing α-pinene SOA for 100
minutes and stabilizing the droplet for 30 minutes, the RH was increased. When the RH reached 76.20%, the fitting error of
the homogeneous Mie algorithm significantly decreased (as shown in Figure 7(a)), indicating a transition from core-shell
morphology to a homogeneous morphology. This demonstrates the capability of our AOT to study the water solubility,
hygroscopicity, and other properties of SOA. Furthermore, these experimental results align with previous studies (Sullivan et
al., 2020), indicating that SOA generated from terpenes tends to form core-shell morphologies during phase separation.
**4 Conclusion**
In this study, we developed and characterized a new single-beam gradient force aerosol AOT system. A customized droplet
particle levitation chamber with a double-floor design was constructed, offering versatility for modifications and enabling
rapid droplet trapping. We conducted a comprehensive characterization and performance assessment of this AOT system. Our
AOT system demonstrated the ability to efficiently capture micron-sized droplets within 30 seconds, significantly improving
capture efficiency. Additionally, the flexibility of the chamber design allowed for adjustments in airflow exchange rate and
direction by altering the shape and size of the air holes in the intermediate part, tailored to specific experimental requirements.
To evaluate the chamber's performance, we trapped NaCl droplets and used the MRFIT algorithm to retrieve their diameter



and refractive index. The experimentally obtained droplet sizes closely matched theoretical values, affirming the chamber's
performance. We also investigated the RH-dependent morphology of droplets, using NaCl droplets mixed with 3-MGA to
measure SRH and phase MRH. Additionally, we generated and added α-pinene and limonene SOA to inorganic droplets in
situ. It is found the formation of a second phase of the droplet occurred, allowing us to study its miscibility and humidity-
dependent morphology. Our findings suggest that the AOT system can effectively study the physical and chemical properties
of typical atmospheric SOA. Our future research using the new AOT system will explore the interaction between secondary
organic matter and various types of trapped droplets, including inorganic salt and organic aerosol droplets.

*Data availability.* The data used in this paper can be obtained from the corresponding author upon request.
*Author contributions.* ZW determined the main goal of this study. XP and YC designed the methods. YC and YM performed
the experiments. XP and YM prepared the paper with contributions from all co-authors. ZX, HL and YS participated the
building of the AOT system. TP provided the codes for WGM analysis and gives comments. ZX and FZ gave comments.
*Competing interests.* At least one of the (co-)authors is a member of the editorial board of Atmospheric Chemistry and Physics.
*Acknowledgments.* The study has been supported by the National Natural Science Foundation of China (grant Nos. 91844301,
42005086), Key Research and Development Program of Zhejiang Province (grant nos. 2021C03165 and 2022C03084), and
the Fundamental Research Funds for the Central Universities (grant no. 2018QNA6008).




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
