# Peer review of "Technical note: Characterization of a single-beam gradient force"

_EGUsphere, 2023_

## Author Comment (AC1)

**Reply to comments on "Technical note: Characterization of a single-beam gradient force aerosol optical tweezer for droplet trapping, phase transitions monitoring, and morphology studies" by Xiangyu Pei et al.**

**Reply to Anonymous Referee #1**

To give an insight into the physicochemical properties of atmospheric aerosols, it is essential to establish effective measurement methods and techniques. However, many techniques currently applied for aerosol measurement cannot acquire information from a single aerosol droplet, such as flow tube and smog chamber, and data acquired from these means are average results of aerosol population, which may omit specific properties of single droplet. In this technical note, the authors presented details of an aerosol optical tweezer-Raman spectroscopy system, and verified its effectiveness in determining characteristics of single aerosol droplet, Generally, I recommend this paper for publication after considering the following comments:

**Response: We truly appreciate the constructive comments by the referee. These comments are very helpful and valuable for improving the quality of our paper. Below we provide a point-by-point response to each comment. The responses are shown in brown and bold fonts, and the added/rewritten parts are presented in blue and bold fonts.**

1. Line 108-109: Humidity in the levitation chamber was maintained by mixing humidified and dry airflow, and the RH results were measured by two sensors at both the inlet and outlet of the chamber. So, I wonder how the authors determined the actual chamber RH, were the final results an average of the two sensors? Since there was a significant difference between the results from the two sensors as shown in Fig.3.

Response: Thank you for the question. Two RH sensors are used to measure RH at two different positions - the chamber inlet and the chamber outlet. The first RH sensor measures the RH after mixing of dry flow and wet flow, before the mixed flow enters the chamber. The second sensor is closed to the droplet trapping position and measures the RH in the lower part of the chamber. Hence the RH measured by the sensor at the outlet of the lower part is used as the actual chamber RH. We have added the following sentence in line 126 in the modified version of the manuscript:

"The RH measured by the sensor at the outlet of the lower part is used as the actual chamber RH, since this sensor is closed to the droplet trapping position and measures the RH in the lower part."

2. Line 131-132: During reactivity experiments, the author declared that the replaceable intermediate part was altered from a flat type to a circular barrier type, can the authors give schematic diagrams of these parts?

Response: Thank you for the comment. The schematic diagrams of the flat type part and the circular barrier type part are shown in Figure S1 (a) and (b), respectively.

[Figure]

**Figure S1. (a) flat type intermediate part; (b) circular barrier type intermediate part**

3. Line 183: The program was developed by who? The reference was missing.

**Response: Sorry for the missed information. The sentence has been modified:**

**"To address this issue and retrieve the diameters and refractive indices for core-shell droplets, we employ another program called Mie Resonance Shell Fitting (MRSFIT), developed by Vennes and Preston (2019)."**

4. Line 220: The position shift of WGMs from ~30 min to ~200 min in Fig.3(c) was not continuous but discrete, can the authors give some explanations?

**Response: This phenomenan usually occurred when RH increased. When RH increased, position of WGMs did not shift, instead, the intensity of WGMs increased continuously and suddenly the WGMs jumped to a larger position. We speculate that when the droplet was trapped by focused laser, the laser will heat the droplet, resulting in a little higher temperature at both places where the laser penetrated the droplet. When RH increased, the water vapor condensed on the droplet surface where temperature was lower, resulting in an ellipsoid shape instead of a sphere. As a result, the average size of the droplet did not change, so position of WGMs did not**

shift. When more and more water vapor condensed on the droplet, the shape of the droplet became more spherical, resulting in forming a better optical cavity and increasing of intensity of WGMs. After adding some amount of water vapor, the average size of the droplet increased significantly and WGMs jumped to a larger position. When RH decreased, the water vapor leaves the droplet surface grandually, the droplet maintained a spherical shape, so position of WGMs shifted to small values continuously.

5. Line 232-233: How much time was needed for the chamber to reach the set RH?

Response: Thank you for the question. In general, the time needed for the chamber to reach the set RH depends on the set RH value. The set RH is higher, the time is longer. For example, from Figure 3(c), we can estimate the time for the chamber to reach the set RH. During the increasing process of RH setting, when RH was changed from 78% to 82%, it took 43 min to reach the set value, whereas when RH was changed from 86% to 90%, it took 73 min to reach the set value. The same phemonemem occurs during the decreasing process of RH setting, when RH was changed from 94% to 90%, it took 63 min to reach the set value; whereas when RH was changed from 82% to 78%, it took 36 min to reach the set value.

---

## Author Comment (AC2)

**Reply to comments on "Technical note: Characterization of a single-beam gradient force aerosol optical tweezer for droplet trapping, phase transitions monitoring, and morphology studies" by Xiangyu Pei et al.**

**Reply to Anonymous Referee #2**

The authors systematically introduced their aerosol optical tweezers coupled with Raman spectroscopy system in a detailed way. They developed a new environmental chamber which has a double-floor and a replaceable intermediate plate, which facilitates stable trapping while introducing different species such as VOC and SOA to the system. In addition to studying the liquid-liquid phase separation (LLPS) of premixed NaCl and 3-methyl glutaric acid (3-MGA) droplets, they were able to in-situ generate and deposit SOA on a NaCl microdroplet and investigate LLPS afterward. Their findings indicate that the AOT system is a powerful tool to in-situ monitor changes in physical and chemical properties of droplets during their evolution under different conditions. I would recommend the paper for publication after addressing the following questions.

**Response: We truly appreciate the constructive comments by the referee. These comments are valuable and helpful for improving the quality of our paper. Below we provide a point-by-point response to each comment. The responses are shown in brown and bold fonts, and the added/rewritten parts are presented in blue and bold fonts.**

Major comment:

The RH is a key parameter in the LLPS measurements, and the authors compared their experimental measurements with theoretical calculations for the relationship between wet particle diameter and RH (Figure 4a). They only provided the data for a short RH range (around 77% to 86%), and it appears that that experiment deviated from theory at higher RH (83%-86% RH). What are the results for a higher RH range, say 86% to 99%?

**Response: Thank you for the question. In Figure 4(a), the RH was measured with the RH sensor at the exit of the chamber, while wet particle diameters were retrieved from the spectra with the Mie fitting program MRFIT. The typical and maximal tolerances for the RH sensor are ±1.5% and ±1.8% when RH is lower than 80%, respectively. When RH increases from 80% to 100%, the typical and maximal tolerances for the RH sensor also increase linearly until ±2% and ±3%, respectively (Sensirion SHT85 datasheet). As a result, the measured RH values at high RH have more uncertainty. In Figure 4a, we think that the experiment deviation from theory is due to the increasing measurement uncertainty of the RH sensor. When RH is higher than 85% during this experiment, the wet particle diameters retrived from the program MRFIT represents hysteresis. For example, when RH is about 86%, the wet particle diameters are in the range of 11.8-12.2 μm, and the range (0.4 μm) is much larger than the value (~0.01 μm) when RH is lower than 86%, which is shown in Figure 4(b). This hysteresis can be explained by that when RH is high, more WGMs peaks represent in each spectrum, the assignment of WGMs labels may have differences since different combination of WGMs labels can lead to the same WGM positions. Considering all the factors above, we still think that the measured values and theoretical values are consistent.**

Minor comments:

OT was introduced as the abbreviation of optical tweezer in Line 46, there is no need to introduce it once more in Line 51.

**Response: It has been changed accordingly.**

Line 55: Change "This results" to this "This result".

**Response: Thanks for the comment. It has been changed accordingly.**

**"This result in a significant enhancement of stimulated Raman scattering signals at specific wavelengths, which are referred to as whispering gallery modes (WGM)."**

Line140: "…to prevent droplet deposition" the surfactant cannot prevent droplet deposition. It helps the deposited droplets to spread on the surface of the coverslip.

**Response: Thanks for the comment. The sentence has been modified into:**

**"Inside the lower part, a glass coverslip is placed and soaked in a surfactant solution (a 50:50 water-to-Decon 90 solution). The surfactant solution is used to help the deposited droplets to spread on the surface of the coverslip."**

Line 183: developed by, something is missing after "by".

**Response: Sorry for the missed information. The sentence has been modified into:**

**"To address this issue and retrieve the diameters and refractive indices for core-shell droplets, we employ another program called Mie Resonance Shell Fitting (MRSFIT), developed by Vennes and Preston (2019)."**

**Reference**

Sensirion SHT 85 datasheet,
https://sensirion.com/media/documents/4B40CEF3/61642381/Sensirion_Humidity_Sensors_SHT85_Datasheet.pdf

---

## Author Response (AR2)

**Reply to comments on "Technical note: Characterization of a single-beam gradient force aerosol optical tweezer for droplet trapping, phase transitions monitoring, and morphology studies" by Xiangyu Pei et al.**

**Reply to Anonymous Referee #2**

Regrarding my major comment, I would recomend to include the data of high RH (86% to 99%) to Figure 4 or SI, with proper discussion of the uncertans.

**Response: Thank you for the comment. We have added the following sentence in line 269 in the modified version of the manuscript:**

**"In this specific experiment, droplets were generated using a 0.4 g/mL NaCl aqueous solution, and the results are presented in Figure 4(a), and the results of another droplet at higher RH are given in the supplement, Figure S1."**

**And we add the following contents in the supplement:**

[Figure]

Figure S1. Comparison between experimental measurements and theoretical calculations of the relationship between wet particle diameter and relative humidity at higher RH compare to Figure 4(a). The lines represent Köhler curves calculated for particles with different diameters: 4.10 µm, 4.20 µm, 4.24 µm, 4.30 µm, and 4.40 µm from left to right, respectively.

Figure S1 shows the comparison between experimental measurements and theoretical calculations of the relationship between wet particle diameter and relative humidity of another droplet at higher RH compare to Figure 4(a). In Figure S1, measured values are generally consistent with the theoretical calculations. In both Figure S1 and Figure 4(a), the RH was measured with the RH sensor at the exit of the chamber, while wet particle diameters were retrieved from the spectra with the Mie fitting program MRFIT. The

typical and maximal tolerances for the RH sensor are ±1.5% and ±1.8% when RH is lower than 80%, respectively. When RH increases from 80% to 100%, the typical and maximal tolerances for the RH sensor also increase linearly until ±2% and ±3%, respectively (Sensirion SHT85 datasheet). As a result, the measured RH values at high RH have more uncertainty. In Figure 4(a) and Figure S1, we think that the experiment deviation from theory is due to the increasing measurement uncertainty of the RH sensor. In Figure 4, when RH is higher than 85% during this experiment, the wet particle diameters retrived from the program MRFIT represents hysteresis. For example, when RH is about 86%, the wet particle diameters are in the range of 11.8-12.2 μm, and the range (0.4 μm) is much larger than the value (~0.01 μm) when RH is lower than 86%, which is shown in Figure 4(b). This hysteresis can be explained by that when RH is high, more WGMs peaks represent in each spectrum, the assignment of WGMs labels may have differences since different combination of WGMs labels can lead to the same WGM positions. In Figure S1, the RH values are in the higher range of 88-94% compare to Figure 4(a), which means the measurement uncertainty of the RH sensor is even higher. Considering all the factors above, we still think that the measured values and theoretical values are generally consistent.

**References**

Sensirion SHT 85 datasheet, https://sensirion.com/media/documents/4B40CEF3/61642381/Sensirion_Humidity_Sensors_SHT85_Datasheet.pdf